# Nutrient limitation determines the fitness of cheaters in bacterial siderophore cooperation

D. Joseph Sexton[1] & Martin Schuster[1]

Cooperative behaviors provide a collective benefit, but are considered costly for the individual. Here, we report that these costs vary dramatically in different contexts and have opposing effects on the selection for non-cooperating cheaters. We investigate a prominent example of bacterial cooperation, the secretion of the peptide siderophore pyoverdine by *Pseudomonas aeruginosa*, under different nutrient-limiting conditions. Using metabolic modeling, we show that pyoverdine incurs a fitness cost only when its building blocks carbon or nitrogen are growth-limiting and are diverted from cellular biomass production. We confirm this result experimentally with a continuous-culture approach. We show that pyoverdine non-producers (cheaters) enjoy a large fitness advantage in co-culture with producers (cooperators) and spread to high frequency when limited by carbon, but not when limited by phosphorus. The principle of nutrient-dependent fitness costs has implications for the stability of cooperation in pathogenic and non-pathogenic environments, in biotechnological applications, and beyond the microbial realm.

[1] Department of Microbiology, Oregon State University, 226 Nash Hall, Corvallis, OR 97331, USA. Correspondence and requests for materials should be addressed to M.S. (email: martin.schuster@oregonstate.edu)

Cooperative behaviors that provide a collective benefit abound across all domains of life, from animals to bacteria[1, 2]. Yet, their evolution and maintenance is difficult to explain given that non-contributing cheaters may reap the benefits of cooperation without incurring the costs[3–5]. Primarily based on studies with animals, several behavioral and ecological factors have been proposed that can provide either direct or indirect benefits to cooperators[5, 6]. Microbial cooperative behaviors, including cell–cell communication, nutrient acquisition, virulence, and biofilm formation, have gained popularity as tractable model systems to experimentally test these factors within existing evolutionary theory[2, 7, 8]. However, relatively little attention has been given to the costs and benefits of these behaviors in the context of microbial growth physiology.

An important ecophysiological parameter that varies widely in nature is resource limitation[9–11]. It has been argued that nutrient limitation generally affects cooperative behavior by increasing its costs, because resources must be diverted away from growth into cooperative functions[12, 13]. Recent work on cooperative secretion in the bacterial pathogen *Pseudomonas aeruginosa* indicates, however, that regulation in response to specific nutrient conditions can minimize these costs[14, 15]. In a regulatory mechanism termed metabolic prudence, a carbon-rich surfactant is produced only when nitrogen is depleted from the growth medium and carbon is in relative excess, preventing non-producing strains from gaining an advantage[14].

Here, we investigate the effects of nutrient limitation in one of the most prominent microbial models of cooperative behavior, iron acquisition via diffusible pyoverdine (PVD) molecules in *P. aeruginosa*[13]. PVDs are a class of non-ribosomal peptide siderophores that chelate iron with high affinity and are produced when iron levels are low[16–18] (Fig. 1a). Iron, an important micronutrient, is often scarce in the environment due to the formation of insoluble oxides under aerobic conditions, chelation by host proteins during infections, or scavenging by competing microbes[19, 20]. Siderophore production qualifies as a cooperative trait. As shown in numerous co-culturing studies, PVD molecules can be shared within a population of cells, benefitting cells other than the focal producer[21–29]. However, the growth advantage of non-producers compared with producers varies among strain pairs, is often marginal, and has only been investigated in complex medium. Consequently, the conditions that may select for the PVD-deficient variants commonly observed in infections and other environments remain unclear[30–33].

We explore the relationship between the fitness cost of PVD production and the type of nutrient limitation using a whole-genome metabolic model paired with controlled in vitro evolution experiments. Modeling reveals two contrasting outcomes: PVD is costly when the limiting nutrient is a building block of PVD, such as carbon (C) or nitrogen (N), suggesting invasion of PVD producers by non-producers. In contrast, PVD is not costly when other nutrients such as iron (Fe), phosphorous (P), or sulfur (S) are limiting as the building blocks are in relative excess, suggesting coexistence of PVD producers and non-producers. We are able to confirm these results with continuous culture chemostats that allow precise adjustment of nutrient availability and require steady production of PVD, but not with batch cultures that have been customary in the field. In sum, we find that it is the intrinsic molecular composition of a secreted product that determines fitness costs and cooperator-cheater dynamics in the context of resource availability. We discuss the significance of this finding for the PVD model system and for social evolution in general.

## Results

**Metabolic modeling.** We initially evaluated the resource-dependent metabolic costs of PVD production in silico, using a

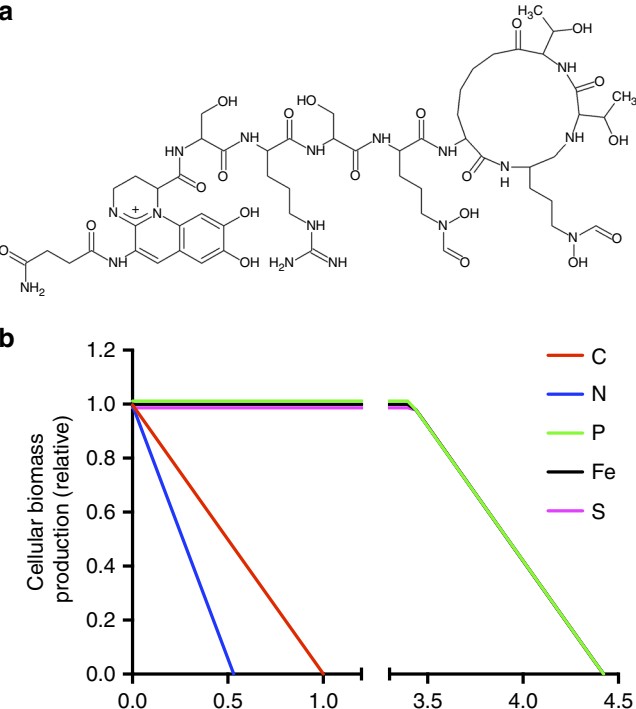

**Fig. 1** Relationship between PVD structure and metabolic cost of secretion. **a** Illustration of the PVD structure (*P. aeruginosa* PAO1 Type I), which is comprised primarily of C and N. **b** Metabolic model. Flux-balance analysis was performed to analyze the trade-off between the levels of cellular biomass production and PVD production. Different nutrient-limiting conditions were achieved by restricting the respective nutrient uptake rates. Graphs for Fe, P, and S-limitation are identical but are offset for visualization purposes

**Table 1 Estimated energetic and biomass requirements for PVD biosynthesis**

| Category | ~P requirement in mmol (g DW)$^{-1}$ | Biomass requirement in g (g DW)$^{-1}$ |
|---|---|---|
| PVD molecule | 11 | 0.56 |
| PVD biosynthesis machinery | 0.26 | 0.0066 |

genome-scale metabolic model of *P. aeruginosa* strain PAO1[34, 35]. This model contains all the basic metabolic pathways in PAO1, but lacks the PVD synthesis reactions. We, therefore, identified all the known biochemical reactions involved in PVD production from the available literature, and added them to the model (Supplementary Table 1). These include accessory reactions, non-ribosomal peptide synthetases, siderophore maturation steps as well as export machinery[17, 18, 36]. In all, 26 high-energy phosphates (~P) are required to produce and secrete 1 molecule of PVD. With PVD concentrations approaching 0.56 g per g of dry weight (DW) in *P. aeruginosa* batch cultures[37], we estimated that approximately 15% of the ATP needed to build a bacterial cell is used for PVD production (see Methods). In terms of both biomass and energy demand, this appears to be a highly costly endeavor. Our model focused on this metabolic aspect and did not separately include synthesis reactions for the PVD biosynthetic enzymes themselves. This was justified as the energy and

biomass requirement for the synthesis of the PVD enzymatic machinery is negligible compared with the synthesis of the PVD molecule from building blocks (Table 1 and Methods).

As a second step, we performed flux-balance analysis to model the relationship between bacterial growth and PVD secretion when different nutrients are growth-limiting. We modeled growth and secretion under aerobic conditions, with glucose as the sole C-source and other macronutrients provided as inorganic salts. Assuming a trade-off in resource allocation between cellular biomass production and secretion, we predicted a negative relationship between the two parameters: Increased PVD secretion should reduce biomass production and vice versa. We indeed found this to be the case under C or N-limiting conditions (Fig. 1b). This is because C and N are constituents of PVD (Fig. 1a), and growth and secretion are, therefore, competing for the same limiting pool of nutrients. We obtained equivalent results with the alternative C-sources glutamate and succinate (data not shown). In contrast, when we limited Fe, P, or S, secretion did not initially affect biomass production as the building blocks of PVD (C and N) were in relative excess. This held true until a secretion threshold was crossed and C again became the growth rate-limiting nutrient. Beyond this threshold, biomass production decreased because the carbon flux required for PVD production exceeded that necessary for cellular biomass production, at a cellular uptake rate set to the maximum possible value. Overall, we identified the relative uptake rates of limiting and non-limiting nutrients as key determinants in the trade-off between growth and secretion (Supplementary Fig. 1).

Further interrogation of our metabolic model revealed that PVD secretion causes the highest flux increases in amino-acid biosynthesis pathways that provide the building blocks for PVD peptide synthesis. Under C and N-limitation, this increase in turn reduces reaction fluxes from central precursor metabolites to cellular biomass. Taken together, we found that fitness cost is not merely a function of the total energetic and biomass requirements, but is also determined by the nature of the growth-rate limiting nutrient. This relationship between nutrient limitation and fitness costs has intriguing implications for the evolutionary stability of siderophore secretion by affecting the dynamics between siderophore producers (cooperators) and non-producers (cheaters) in a mixed population.

**Batch culture growth experiments**. To empirically test the predictions of our metabolic model, we sought to compare the fitness of a PVD-producing strain to a non-producing strain when different nutrients limit growth. According to our model, PVD production is most costly when a building block of the siderophore is growth-limiting (Fig. 1). Hence, a non-producer should enjoy the greatest relative fitness benefit under these conditions. We constructed an isogenic pvdS gene (PA2426) deletion mutant in the background of the PAO1 parent that served as a non-producing strain. This gene encodes an alternative sigma factor that regulates transcription of a set of genes required for the synthesis of PVD[16, 38]. The deletion of pvdS is ecologically relevant because clinical PVD-deficient P. aeruginosa isolates are often mutated in this gene[30, 39, 40].

For growth experiments, we chose to focus on media limited by either C or P; each representing conditions where we predict PVD production to come at a high or low fitness cost, respectively. We started with a defined, synthetic medium derived from Evan's minimal medium[41], referred to herein as modified Evan's medium (MEM). MEM was limited by either glucose as the sole C-source (C-MEM) or phosphate as the sole P-source (P-MEM). Both media were supplemented with ethylenediamine-di (o-hydroxyphenylacetic acid) (EDDHA), a strong $Fe^{3+}$ chelator

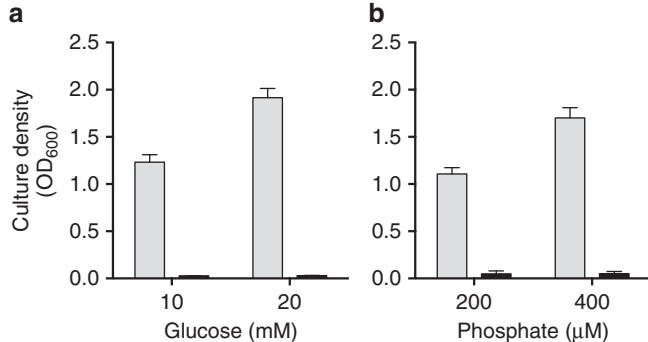

**Fig. 2** Growth yield of P. aeruginosa in low-iron batch cultures. Individual cultures of the WT strain (gray bars) or the pvdS mutant (black bars) were grown in MEM for 24 h, and cell densities were measured as optical density at 600 nm ($OD_{600}$). The initial concentrations of glucose and phosphate in the medium varied as follows. In **a**, glucose varied while phosphate was held constant at 10 mM. In **b**, phosphate varied while glucose was held at 20 mM. The medium also contained 1 μM Fe and 40 μM of the chelator EDDHA that restricted growth of the pvdS mutant. Error bars show standard error of the mean (SEM), n = 3

that requires PVD for growth[38]. A second, low-affinity siderophore produced by P. aeruginosa, pyochelin, is insufficient for growth in the presence of this chelator[38]. The Fe concentration was set at 1 μM, which is low enough to stimulate PVD production but high enough to not limit the growth yield set by C or P concentrations. We chose C and P levels as well as cultivation times such that the WT reached virtually identical and saturating final cell densities in both media (Fig. 2 and Supplementary Fig. 2a). We verified that the intended nutrient was limiting by doubling its concentration and observing a nearly proportional increase in the culture density of the WT (Fig. 2). The pvdS mutant, in contrast, was unable to grow by itself under these conditions and did not produce any PVD (Fig. 2 and Supplementary Fig. 2). To overcome this growth defect in the pre-cultures used as inocula, we formulated an Fe-replete MEM that allowed the mutant to grow as well as the WT (Supplementary Fig. 3).

We then initiated batch culture competition experiments to compare the fitness of the pvdS mutant relative to the WT in C-MEM and P-MEM (Fig. 3a). In both cases the frequency of the mutant remained constant throughout the duration of co-culture. The frequency of the mutant also did not change in C-MEM after three additional subcultures (average relative fitness $w = 1.07$ after 25 generations, standard error of the mean (SEM) = 0.06, n = 4; no significant difference from 1 by one-sample t-test, p = 0.33). This shows that the pvdS mutant is indeed capable of utilizing the PVD produced by the WT, but it does not enjoy a relative growth advantage, regardless of the limiting nutrient. This result was similar to a variety of co-culturing experiments we had conducted (Supplementary Note 1, Supplementary Fig. 4 and Supplementary Table 2), which included conditions similar to those used in published studies[21, 23, 25, 26, 29, 33, 42–44]. Collectively, we found that PVD non-producers grow at equal rates, but never faster, than the WT. This finding is inconsistent with the notion that non-producing cheater mutants can invade a producing population by avoiding the cost of cooperation, at least under the conditions explored thus far.

These results led us to carefully reconsider the use of a batch culture format, which has inherent limitations regardless of the media type used. Batch cultures are frequently used for the growth of microorganisms due to their low cost and ease of operation. However, the boom-bust pattern of growth alters the

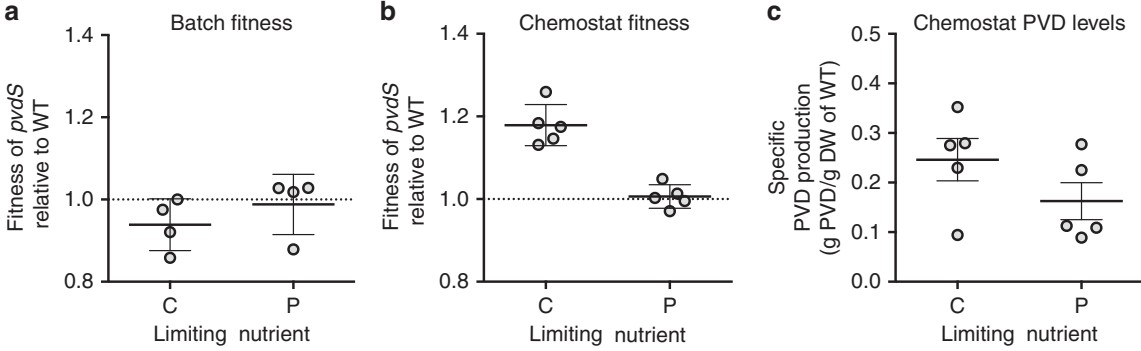

**Fig. 3** Fitness costs of PVD production according to growth-limiting nutrient and culturing format. **a**, **b** Relative fitness of the *P. aeruginosa pvdS* mutant in **a** batch culture and in **b** chemostat culture. Co-cultures of the WT and the *pvdS* mutant were initiated at a 1:1 ratio in either C-MEM or P-MEM. The duration of growth was 24 h in batch and 6 days in chemostat culture. Relative fitness *w* was calculated as the ratio of average growth rates, considering initial and final strain frequencies ($w > 1$ indicates an increase in mutant frequency, $w < 1$ indicates a decrease in mutant frequency, and $w = 0$ indicates no change in mutant frequency over time). Relative fitness is significantly different from 1 only in C-MEM chemostat culture, but not in C-MEM batch, P-MEM batch, and in P-MEM chemostat cultures (one-sample *t*-test; $p = 0.0013$, 0.15, 0.77, and 0.65, respectively). Relative fitness in C-MEM is significantly different than in P-MEM in chemostat culture but not in batch culture (two-sample *t*-test, $p = 0.00010$ and 0.34, respectively). **c** Specific PVD production levels in chemostat culture. Shown are the average amounts produced by the WT subpopulation over time. PVD levels in C-MEM were not significantly different than in P-MEM (two-sample *t*-test, $p = 0.18$). Each individual data point represents a separate biological replicate. *Error bars* indicate SEM

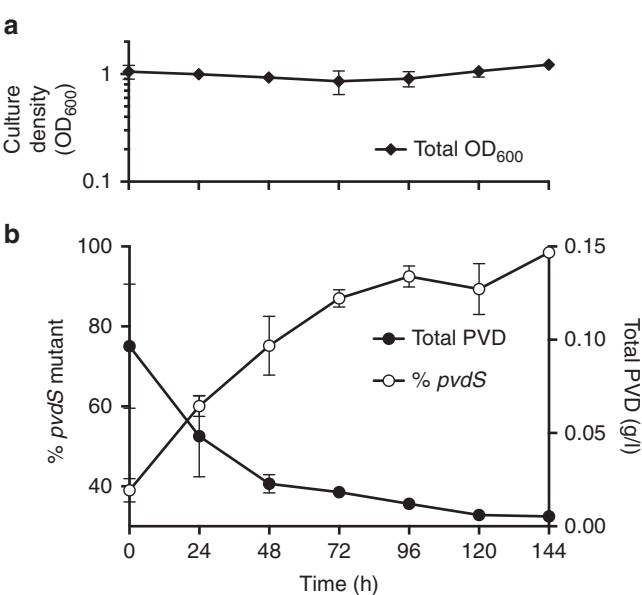

**Fig. 4** Change in population density, frequency and PVD concentration during C-limited chemostat culture. **a** Total population density. **b** Proportion of the *pvdS* mutant in WT co-culture, and total concentration of PVD in the chemostat over time. Co-cultures of WT and *pvdS* mutant were initiated in C-MEM at the indicated ratio. *Error bars* indicate SEM, $n = 5$

media composition over time, causing changes in the growth-rate limiting nutrient and causing accumulation of secreted metabolites, including PVD itself[45]. This dynamic growth environment of batch cultures and consequently, the changing selective pressures experienced by bacterial populations, led us to pursue an alternative culturing approach with a higher degree of control.

**Chemostat culture experiments.** We eliminated the uncertainties associated with batch culture by turning to a chemostat format, an open system where fresh nutrients continuously displace spent medium. In this system, both cell density and growth rate are held constant, as determined by the concentration of the growth-rate limiting nutrient and the flow rate of

the growth medium, respectively[41]. Furthermore, chemostat culturing ensures continuous PVD production from the WT, as the constant flow of medium limits PVD accumulation. Using this approach, we reasoned that the fitness cost associated with PVD production in relation to the limiting nutrient could be more clearly resolved. We again set out to compare the fitness of the *pvdS* mutant relative to the WT in C-MEM and P-MEM. Chemostats were operated at a dilution rate of $0.2\,h^{-1}$ (corresponding to a generation time of 3.5 h), and sampled daily to determine strain frequencies and PVD concentrations.

We found that the *pvdS* mutant did indeed enjoy a significant relative fitness advantage in C-MEM, but not in P-MEM (Fig. 3b). This finding confirmed our prediction that the fitness cost of PVD production is contingent on the limiting nutrient. Importantly, we found that PVD secretion levels were comparable in both growth media, excluding the possibility that observed differences in relative fitness are merely a function of secretion activity (Fig. 3c). The measured PVD levels are somewhat lower than those cited in Table 1, but are still predicted to incur costs that are substantially above those for PVD synthesis machinery.

The growth advantage of the *pvdS* mutant led to rapid invasion of the population, approaching an apparent equilibrium at a *pvdS* mutant frequency of ~95 % (Fig. 4). The very high cheater tolerance in the C-limited chemostat without any reduction in total cell density (Fig. 4a) indicates that PVD secretion levels by the WT far exceed those necessary to sustain its own growth. The mechanism that may limit further cheater invasion is currently not clear, but it is evident that the increase in cheater frequency correlates with a decrease in the total PVD concentration present in the chemostat (Fig. 4b). This substantial decrease in initially excessive PVD levels reduces Fe availability and hence could cause a change in the limiting nutrient from C to Fe. Fe limitation, as predicted by modeling above, would again stabilize PVD cooperation.

In order to relate experimental data to those from our metabolic model, we further determined growth and PVD secretion rates during the early *pvdS* mutant enrichment phase in C-limited conditions (up to 72 h of culturing), based on the measured dynamics of subpopulation densities and PVD concentrations. The growth rates of the WT and *pvdS* mutant are $0.181 \pm 0.003\,h^{-1}$ and $0.210 \pm 0.002\,h^{-1}$, respectively, yielding a 13.8% reduction in growth attributable to PVD secretion. The

PVD secretion rate of the WT is $0.044 \pm 0.011\,\mathrm{g\,(g\,DW\,h)^{-1}}$. Reinterrogating our metabolic model, we find that a PVD secretion rate of this magnitude reduces the growth rate from $0.210\,\mathrm{h^{-1}}$ to $0.184\,\mathrm{h^{-1}}$ under C-limitation, amounting to a 12.4% reduction. Thus, there is very good agreement between the two approaches.

## Discussion

PVD production by *Pseudomonas* is one of the main microbial models for the evolution of cooperation. Because the costs of cooperation are an important determinant for its evolutionary stability, a detailed understanding of the environmental and physiological conditions that affect these costs is critical. Cost-benefit considerations seem particularly important in shaping the evolution of cooperative secretions that are lost to the environment[46–48].

In this study, we combined metabolic modeling with defined culturing experiments to demonstrate that the growth-limiting nutrient profoundly impacts the fitness costs and stability of PVD secretion in *P. aeruginosa*. When cell growth is limited by P, S or Fe, PVD production does not translate into a fitness cost because the resources needed to make the siderophore are in relative excess. In contrast, when C or N is growth-limiting, PVD production competes with cellular biomass synthesis for the same limited resource, resulting in a significant fitness cost. We experimentally verified our modeling predictions with a chemostat system. PVD production came at a fitness cost and allowed invasion of non-producing cheaters when C but not P was limiting, despite similar PVD secretion levels (Fig. 3b, c).

It is considered an established principle in microbial physiology that cells tightly couple biomass yield with ATP generation when limited by the C and energy source[49]. In contrast, when limited by a nutrient other than C and energy source, cells tend to redirect the non-limiting cellular C flux to other functions, including secretion[11, 49]. Our results indicate that when secretion is nevertheless required under C limitation, a trade-off in nutrient allocation between growth and secretion is inevitable. P-limiting conditions in particular have been associated with low levels of phosphorylated compounds, including ATP[50, 51]. In our case, however, ATP did not appear to be the growth-limiting intracellular metabolite because PVD production, along with its ATP demand, did not result in a relative growth disadvantage.

Furthermore, we found that PVD-producers are able to sustain a very high "cheater-load" of non-producers at an apparent equilibrium frequency in C-limited chemostat co-culture (Fig. 4). This result is remarkable. In a well-mixed environment without assortment, a costly cooperative behavior should not be evolutionarily stable[3–5]. Non-producing cheaters are expected to invade a population of cooperators and cause its collapse[52, 53]. We offer two explanations for our observation. First, as mentioned above, nutrient-dependent fitness costs could shift over time. Decreasing PVD levels could change growth conditions from C-limited to de-facto Fe-limited. Under these new conditions, PVD secretion would incur no costs as C and N are in relative excess. Second, *P. aeruginosa* PVD may not be a fully secreted product but may instead be partially retained by the producing cell[54, 55]. This property would be of increased significance at low producer frequencies when extracellular, secreted PVD levels available to non-producers are also low. In fact, retaining as little as 1% of a secreted product can favor cooperation in a way that leads to an equilibrium between producers and non-producers[56].

Our findings have important implications for the use of PVD cooperation as a microbial model system for social evolution. The C-limited chemostat proved to be a unique condition where the non-producer had a fitness advantage and was able to invade a PVD-producing population, contrasting the P-limited chemostat and a wide collection of culturing permutations we explored in a batch culture format (Fig. 3a, Supplementary Fig. 4, and Supplementary Table 2). After comparing our findings with previous studies, we are led to conclude that the use of undefined media in a batch culture format, an approach thematic to published literature, does not provide a consistent selective advantage for non-producers: Cheater invasion (i.e., a relative fitness > 1) is observed in some cases but is often marginal, and is less pronounced with isogenic than with non-isogenic strain pairs (see Supplementary Note 1)[21, 23–26, 29, 42, 43].

A first step we have taken here toward clearly defining growth conditions was the use of a synthetic medium. Heated debates have emerged in recent years regarding the selective conditions created by CAA-based medium, as well as another commonly used undefined medium known as King's B[33, 44]. This dispute challenges the interpretation of an entire collection of social evolution studies on siderophores. Exacerbating this issue further, distinct PVD phenotypes have been observed in CAA preparations originating from different manufactures, suggesting efforts to characterize the attributes of this medium may be chasing a moving target[44].

A second step we have taken was the use of a continuous-culture chemostat system. As we demonstrated, the relationship between the limiting nutrient and the fitness cost of PVD production was obscured in our batch culture experiments (Fig. 3a). There are two plausible explanations for this. The first relates to the dynamic changes in media composition due to the influence of bacterial growth physiology[45]. This fact is relevant in our context, as we could only confirm the intended limiting nutrient was the first to be depleted (Fig. 2). It is likely that Fe is the limiting nutrient early in growth, as de-novo PVD production is required to scavenge iron from the strong chelator EDDHA. The second explanation considers the ability of *Pseudomonas* to recycle PVD and adapt its phenotype accordingly. Indeed, after Fe is unloaded into the periplasm, PVD is returned to the extracellular environment and can be reused many times[36]. As PVD accumulates, the producing strain reduces costs by suppressing PVD biosynthesis and benefits from the PVD produced by previous generations. This intrinsic property has been suggested to affect the evolutionary stability of PVD cooperation[43]. Although both explanations have merit, the very nature of the batch culture format makes it difficult to distinguish the two. Our continuous-culture approach avoids these uncertainties as it provides the ability to hold a known selective pressure constant, a feature that is desirable for experimental evolution studies in general[45].

Distinguishing the conditions that select for secretion deficiency or instead stabilize secretion remains a challenging yet important goal. Microbial secretions have an important role in natural microbial communities, in pathogenesis and agriculture, in biotechnological applications, and in antimicrobial intervention[57–61]. Medical relevance is apparent in the chronic lung infections of cystic fibrosis patients, where secretion-deficient strains of *P. aeruginosa* are consistently isolated[30, 32, 40]. Recent evidence suggests that PVD-negative strains evolved as social cheaters, because they tend to retain their PVD receptor when cross-feeding on a co-infecting PVD-producing strain is possible[30]. However, our data indicate that the ability to cross-feed is not alone sufficient to conclude that non-producers have a fitness advantage, even when siderophores are required for growth (Fig. 3, Supplementary Fig. 4, and Supplementary Table 2). This highlights the challenges to understanding social dynamics in natural environments and emphasizes the need to better characterize the selective pressures in a given context.

The principles we have demonstrated in vitro provide a framework to consider the evolutionary origin of secretion-deficient isolates observed in natural environments. By pairing knowledge about the composition of a given secreted product with local resource availability, general predictions can be made. This approach can be illustrated by considering marine ecosystems, where many strains have been isolated that require siderophores for growth but do not produce them[31, 62]. Our findings suggest peptide siderophores should be particularly costly when N is limiting, which is often characteristic of low latitude surface waters with minimal nutrient inputs[9]. In contrast, Fe can be limiting in sub-surface waters when upwelling or other sources increase nutrient concentrations beyond what can be fixed from the atmosphere[9]. Here production may come at little to no cost, due to the relative excess of carbon and nitrogen. Other features which impact the stability of production, such as spatial structure and siderophore durability, should not be neglected in aquatic environments, even though they are expected to play stronger roles in highly structured environments, such as in soil for example[63]. The principle of nutrient-dependent fitness costs can be used in an analogous fashion to devise culture conditions that stabilize the production of secreted enzymes or metabolites in biotechnological applications.

Taken together, our results exemplify how the relationship between resource availability and the molecular properties of the secreted product itself impact the stability of cooperation. The limiting resource is likely to impact the stability of many social behaviors that involve shared products, generally referred to as public goods. In *P. aeruginosa*, it has already been shown that quorum-sensing dependent secretions are "prudently" regulated in a way that ensures sufficient supply of building blocks[14, 15]. This suggests that the regulatory network has adapted to minimize the fitness cost of public goods production by responding to the available resources. Although we did not focus our study on the regulation of PVD expression, it is known that production is stimulated when environmental Fe concentrations are low[16]. This alone may be sufficient in many cases to prevent cheating, namely if Fe levels are so low that they are growth-limiting. Consequently, the PVD building blocks C and N are in relative excess such that PVD production comes at no cost. In general, our work highlights the importance of ecophysiological context for the evolution and maintenance of cooperative behavior. Fitness costs may vary more widely and social interactions may be more dynamic than currently appreciated.

## Methods

**Metabolic model.** For flux-balance analysis, we used the whole-genome metabolic model of *P. aeruginosa*, iMO1086, constructed by Oberhardt et al.[34, 35]. PVD synthesis reactions were added to the metabolic model as described in Supplemental Table 1. All computations were performed with the COBRA toolbox in Matlab (version R2013b, Mathworks, Natick MA)[64]. To assess the trade-off between growth (i.e., cellular biomass production) and PVD secretion as shown in Fig. 1b, we performed a robustness analysis that calculates the growth rate as a function of the PVD secretion rate. C, N, P, S, or Fe limitation was achieved by constraining the respective nutrient uptake rate such that the resulting growth rate in the absence of any PVD secretion is identical to the experimental value of $0.2\,h^{-1}$. The uptake rates of the non-limiting nutrients remained effectively unconstrained, with the exception of the C (glucose) uptake rate, which was set to the maximally possible rate of $12\,mmol\,(g\,DW\,h)^{-1}$, as experimentally determined[65]. For graphing, the growth rate in the absence of secretion ($0.2\,h^{-1}$) and the PVD secretion rate that yields zero growth under C-limitation ($0.25\,mmol\,[g\,DW\,h]^{-1}$, equal to $0.34\,g\,[g\,DW\,h]^{-1}$) were normalized to one. Different, physiologically relevant C-sources[15] were tested by constraining the uptake rates for glutamate or succinate instead of glucose.

To examine the effect of PVD secretion on the relative reduction in growth rate under a range of different C and P levels, we decreased the respective uptake rates in twofold increments, starting with the rates that permit maximal growth in the absence of PVD secretion (C uptake rate of $12\,mmol\,[g\,DW\,h]^{-1}$ and P uptake rate of $0.8\,mmol\,[g\,DW\,h]^{-1}$).

To investigate the metabolic basis for the trade-off between growth and secretion, we determined the ratios of reaction fluxes in the presence vs. the absence of PVD secretion (growth rate of $0.2\,h^{-1}$ and PVD secretion rate of $0.05\,mmol\,(g\,DW\,h)^{-1}$, equal to $0.068\,g\,(g\,DW\,h)^{-1}$, and sorted them by magnitude.

**Estimation of energetic and biomass requirements for PVD production.** We estimated the requirements for PVD biosynthesis, as presented in the Results, in terms of high-energy phosphates (mol ~P per g DW)[66] and in terms of biomass (g PVD per g DW). We separately considered PVD molecule biosynthesis, and manufacturing of the PVD synthesis machinery. We estimated energetic requirements from building blocks, and we assumed that the requirements of cellular synthesis in *P. aeruginosa* are similar to those in *E. coli*, for which most values have been determined[67]. In minimal medium with good aeration, 1 g of culture produces 0.56 g of PVD[37], equal to 0.41 mmol ($Mr = 1350$ for PVD Type I). To estimate the energetic requirement for PVD molecule synthesis from building blocks, we considered that, according to our calculation, 1 molecule PVD demands 26 ~P. The energetic requirement is then 0.41 mmol PVD $(g\,DW)^{-1} \times 26$ mol ~P (mol PVD)$^{-1} = 11$ mmol ~P $(g\,DW)^{-1}$.

To approximate the requirement of building the PVD synthesis machinery, we considered that the energy demand for the production of total protein in 1 g of cells is 22 mmol ~P $(g\,DW)^{-1}$ (ref. [66]). Without any information about expression levels of PVD machinery, we assumed that the contribution of PVD protein biosynthesis is equal to the length of the coding sequence for annotated PVD biosynthesis genes (69,729 bp) in relation to the length of the entire coding sequence in the *P. aeruginosa* genome (90% of the total 6.26 Mbp sequence)[68, 69]. We could then estimate that the requirement for the PVD synthesis machinery is 22 mmol ~P $(g\,DW)^{-1} \times 1.2\% = 0.26$ mmol ~P $(g\,DW)^{-1}$. We further related these results to total cellular biosynthesis, which requires 72 mmol ~P to generate 1 g DW of *E. coli* cells under aerobic conditions, in minimal medium with glucose as the sole C-source[67].

We finally estimated the biomass requirement for the PVD biosynthesis machinery. Given that about 55% of cell DW is protein[66], the respective biomass requirement is 1 g DW $\times 55\%$ $(g\,DW)^{-1} \times 1.2\% = 6.6$ mg $(g\,DW)^{-1}$.

**Bacterial strains and growth media.** We used the PVD-producing *P. aeruginosa* WT strain PAO1 (ATCC 15692) and a derived, non-producing *pvdS* in-frame deletion mutant, which was constructed in the same genetic background by splicing-by-overlap-extension PCR and allelic exchange as described[70]. The *pvdS* gene deletion was generated using 5'-N6GAGCTCGACATCGTTTTCGGC GGGCGGG-3' and 5'-N6TCTAGAGATGCTGGCGCGCTCGGGCATCCAG-3' as flanking primers and 5'-TTCCGACATGGAAATCACCTTGCTGCGGAG-3' and 5'-AGCAAGGTGATTTCCATGTCGGAAGCCCGCCGCTGACGGCGGC-GAGCATTCCTCA-3' as overlapping internal primers. Restriction enzyme sites (*Sac*I and *Xba*I, respectively) for cloning into an allelic exchange vector are underlined.

Cultures were routinely maintained in Lennox LB liquid medium or on Lennox LB agar plates at 37 °C. The synthetic minimal medium used for batch culture and chemostat experiments is a modification of Evan's minimal medium (MEM)[41]. The base salt composition is 10 mM KCl, 1.25 mM MgCl$_2$·6H$_2$O, 50 mM NH$_4$Cl, 2 mM Na$_2$SO$_4$, 20 μM CaCl$_2$·2H$_2$O, 50 mM 3-(N-morpholino) propanesulfonic acid (pH = 7), 10 μM ZnO, 20 μM MnCl$_2$·4 H$_2$O, 2 μM CuCl$_2$·2H$_2$O, 4 μM CoCl$_2$·6H$_2$O, 2 μM H$_3$BO$_3$, 33 nM Na$_2$MoO$_4$·2H$_2$O, 1 μM FeCl$_3$·6H$_2$O. Fe was chelated by adding 40 μM EDDHA. C-MEM had 10 mM D-glucose and 10 mM NaH$_2$PO$_4$. P-MEM had 20 mM D-glucose and 200 μM NaH$_2$PO$_4$. Before initiating experiments in C-MEM or P-MEM, single colonies were used to inoculate an unchelated, Fe-replete medium of otherwise identical salt composition with 20 mM D-glucose, 10 mM NaH$_2$PO$_4$ and 50 μM FeCl$_3$·6H$_2$O. This provided a culturing medium in which both the WT and the *pvdS* mutant could grow independently. Cells were washed twice in MEM lacking EDDHA, D-glucose, NaH$_2$PO$_4$ and FeCl$_3$·6H$_2$O before sub-culturing into C-MEM or P-MEM. Culture densities were routinely monitored by culture optical density at 600 nm (OD$_{600}$).

**Batch culture experiments.** Batch cultures were grown at 37 °C with shaking. To demonstrate the growth-limiting nutrient, mono-culture growth experiments with WT and *pvdS* mutant were performed in MEM with varying glucose or phosphate concentrations. Cultures were inoculated at an OD$_{600}$ of 0.01. Growth was measured as OD$_{600}$ after 24 h of cultivation. For time courses, growth and PVD production were measured in 6 h intervals. For co-culture competitions, the WT and *pvdS* mutant were mixed to 1:1 ratios and diluted to an OD$_{600}$ of 0.01 in either C-MEM or P-MEM. Cultures were sampled at 0 and 24 h to measure strain frequencies and calculate fitness values for each of the four biological replicates. In the case of successive subculturing, C-MEM co-cultures were diluted into fresh medium and incubated again for 24 h, for a total of four growth cycles.

**Chemostat operation.** Chemostats were built in-house as described previously, with minor modifications[15]. They were operated at 37 °C, well aerated, stirred, and maintained at volumes of 100 ml. Cultures of the WT and *pvdS* mutant were mixed to 1:1 ratios, inoculated to an OD$_{600}$ of 0.05 and grown in batch mode until reaching an OD$_{600}$ of 1 in late exponential phase. At this point, chemostat mode

was initiated. Fresh culture medium was pumped into the culture vessel at a dilution rate ($D$) of 0.2 h$^{-1}$ using a peristaltic pump. Chemostats were sampled every 24 h to determine culture densities, PVD concentrations and frequencies of strain populations. A total of five biological replicates were performed.

**Strain frequencies and relative fitness.** The relative frequencies of WT and *pvdS* mutant sub-populations were distinguished by patching 100 randomly selected colonies onto King's B agar, which stimulates production of PVD[71]. Strains were identified as PVD producers or non-producers based on a distinct green or white phenotype, respectively[71]. Relative fitness ($w$) of non-producers was subsequently calculated as the ratio of average growth rates or Malthusian parameters ($\mu_{pvds}/\mu_{WT}$) as described[72]. Calculation of the average growth rate $\mu$ for each sub-population considers the initial and final sub-population densities $N_0$ and $N_1$, respectively, and the culturing time $t$. For batch cultures, $\mu = \ln(N_1/N_0)/t$ (with $t = 24$ h), and for chemostat cultures, $\mu = \ln(N_1/N_0)/t + D$, which considers the dilution rate (with $t = 144$ h).

**Specific PVD production.** PVD concentrations were estimated from culture supernatants. Relative fluorescence units (excitation and emission wavelengths of 400 and 460 nm, respectively) were measured with a TECAN Infinite 200 plate reader (Tecan Group Ltd, Männedorf, Switzerland). PVD concentrations were interpolated from a standard curve generated with purified PVD$_{PAO1}$ (EMC Micro-collections, Tübingen, Germany). For data reported in Fig. 3c, the obtained, values were divided by the subpopulation density of the PVD-producing WT measured at the same time point. The resulting specific PVD production level is expressed as g per g DW of WT. DW of cells was quantified by operating dedicated chemostats under conditions identical to those used for population and PVD measurements. This approach avoided perturbations from the withdrawal of large sample volumes needed for DW measurements. Based on these measurements, an OD-to-DW conversion factor was calculated (OD$_{600}$ of 1 equals 0.48 g DW l$^{-1}$). For each biological replicate, specific PVD production values were determined at 24 h intervals. Data presented show the average specific PVD production level for each replicate.

**Quantitative comparison of metabolic model and chemostat experiment.** As a first step, we determined specific growth and PVD secretion rates in the C-limited chemostat, only considering the *pvdS* mutant enrichment phase (0 to 72 h) before an apparent equilibrium was reached. A growth rate $\mu$ for the WT and *pvdS* mutant subpopulations was determined by fitting replicate time courses of subpopulation densities to a standard growth rate equation of the form $N_t = N_0 \ e^{(\mu - D) \ t}$ using linear regression. $N_t$ and $N_0$ denote subpopulation densities at time $t$ and at time zero, respectively. A PVD secretion rate $q$ was determined by fitting replicate time courses of PVD concentrations to a model that considers the change in PVD concentrations and the WT growth rate during the *pvdS* mutant enrichment phase. Changes in the PVD concentrations over time are given by the differential equation $dPVD/dt = q \ N_t - D \ PVD_t$. Integration and incorporation of the growth rate equation yields $PVD_t = (PVD_0 - N_0 \ q/\mu_{WT}) \ e^{-D \ t} + N_0 \ q/\mu_{WT} \ e^{-(\mu - D) \ t}$, with $PVD_t$ as PVD concentrations at time $t$ and $PVD_0$ as initial PVD concentration. Graphpad Prism software was used for linear and non-linear regression analysis. Rates $\mu$ and $q$ are reported with SEM in the main text.

In a second step, we matched empirical data to modeling data. In the model, we set the C uptake rate such that the growth rate, in the absence of any secretion, is identical to the experimentally determined growth rate of the *pvdS* mutant. Next, we modeled PVD secretion under C-limiting conditions at the experimentally determined PVD secretion rate of the WT, and we determined the corresponding reduction in growth rate.

**Data availability.** All relevant data are available within the article and its supplementary information, or from the authors upon request.

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

## Acknowledgements

We thank Radhakrishnan Mahadevan from the University of Toronto and Ganti Murthy from Oregon State University for their help with metabolic modeling. We thank Katja Bettenbrock, Steffen Klamt, Ruxandra Rehner, and Helga Tietgens from the Max Planck Institute Magdeburg for sharing their expertise on continuous culture systems. We also thank Amandip Singh for experimental assistance. This study was funded by a grant from the National Science Foundation (no. 1652837) and by an Alexander von Humboldt Fellowship for Experienced Researchers (both to M.S.).

## Author contributions

M.S. conceived of the study and D.J.S. performed experiments. D.J.S. and M.S. both designed experiments, analyzed data, and wrote the manuscript.

## Additional information

**Competing interests:** The authors declare no competing financial interests.

