## [Peer Review file · Nature Communications]

Reviewers' comments:

Reviewer #1 (Remarks to the Author):

In this manuscript, Sexton and Schuster investigate, from a metabolic perspective, the cost of a secreted "common goods" molecule (an iron scavenging siderophore) by *P. aeruginosa*. For this, they use a genome-scale metabolic modeling approach followed by chemostat cultivations (allowing constant growth conditions with a single limiting nutrient) with a wild-type strain and a siderophore production deficient strain. Overall, they conclude that the metabolic cost of the siderophore production is dependent on the limiting nutrient (or rather excess availability of specific nutrients) and this ultimately can regulate the emergence and stability of the cheater population. They discuss this conclusion in an ecological context of *P. aeruginosa*. The conclusion is of general interest. Although the study is sound on technical side, there are several missing links in the chain connecting the data to conclusions.

Major comments:

1. The title is misleading, as the study does not address "evolutionary stability". In fact, the chemostats have not been run long enough to see evolution (e.g. spontaneous emergence of cheaters through mutations). The study is still interesting but the title and other related phrases should be rewritten to avoid misunderstanding.
2. The link between the modeling and the experiments is there but it is not quantitative. The qualitative hypothesis that the cost is high when substrates needed for the siderophore production are limiting is obvious and could be easily put forward / proved without any modeling. The value of the model would lie in predicting the actual cost in a quantitative manner. This cost is not tested/matched against the experimental observations.
3. The "surprising" result of high cheater invasion (Fig 4B) is not so surprising but rather "interesting". One of course cannot observe what is infeasible! Again, the authors should be able to show this by using a simple two-component population model. The value here would be if the model could predict this stability point. (Similar arguments would apply to the comments on line 16-17 on page 8).
4. Page 10 line 5: hypothesis of C and N excess is easily testable in chemostats and should be done to pin down what is happening.
5. Page 7 line 10-11: Here I see again a conceptual problem. The proportion of cheaters will of course never be more than what the non-cheaters can support. So this comment on inconsistency is violating the fundamental theory of selection.
6. Abstract line 1: "but are generally considered costly for the individual". By definition, cooperation should cost individual. The question is never about cost but the benefit/cost ratio.
7. Page 5 line 23: "we found.." should actually be 'we found, AS EXPECTED, ...' since there is nothing else that one could expect.
8. Page 9 line 22: PVD production will not have effect on rate in a chemostat but will have effect on yield.
7. Discussion page 13 line 1: the data does not show (or there is no reference to other literature) whether it is regulatory network acting to shift the siderophore production levels (e.g. by altering gene expression) or it is the metabolic flux/kinetic control subject to intracellular metabolite levels.

Reviewer #2 (Remarks to the Author):

The central questions addressed is how nutrient limitation affects cooperative behavior in a microbial community. Addressing this question is of tremendous significance for fundamental evolutionary theory, with applications in a variety of areas.

The paper capitalizes on the use of mathematical modeling and controlled in vitro evolution experiments to address this complex, systems-level question.

It'd be interesting to make more use of the model in answering some of the questions posed in the manuscript. Which metabolic pathways are potentially active in the various conditions? Might the model be used to understand why there is a high cheater tolerance in the c-limited chemostat without reducing the total cell density? Could the model inform why the pvd mutant has a fitness advantage in the c-mem vs p-mem?

Related to the point above, the "parsimonious" explanation with respect to the chemical composition of PVD could be elaborated on more directly. Are there alternative pathways for which biomass or PVD production can be maintained in a more rich media but which are "unavailable" in the c-limited or n-limited media?

There are several statements made in the manuscript that would benefit from a citation (e.g., from the discussion "an established principle...that cells tightly couple biomass yield with ATP generation when limited by the C and energy source").

The methods state that "growth rate" is calculated and compared to the PVD secretion rate. How are these rates normalized?

The methods state that the "C,N,P,S, and Fe uptake rates were set such that the resulting growth rate in the absence of any PVD secretion is identical to the experimental value"...but wouldn't the relative balance between the C,N,P,S, and Fe uptake rates affect some of the analyses comparing growth and PVD secretion rate? Perhaps an analysis varying the relative uptake rates of those inputs should be done to verify the observations of the balance between growth and PVD secretion.

How are the "C,N,P,S, and Fe" uptake rates actually constrained? It looks like "C" is really defined as glucose...but do the results change when "C" is some other carbon source?

Fig 3 would benefit from a title above A and B to readily distinguish between batch and chemostat.

Reviewer #3 (Remarks to the Author):

Summary:

The authors study the underlying costs of producing the well-studied public good molecule, the iron-chelating pyoverdine siderophore, produced by *Pseudomonas aeruginosa* bacteria. They show how production costs vary in different environmental conditions and determine the implications this has on the evolutionary stability of cooperation. Using metabolic modelling and experiments, they show that although the production of pyoverdine is always energetically costly, it only has a fitness cost when the essential nutrients required to make it (carbon and nitrogen) are limiting. The metabolic model shows that there is an inverse relationship between pyoverdine production and cell biomass only when C and N are limited but not when Fe, P or S are limited. In contrast to predictions made by the model, batch culture experiments show that there is no relative growth advantage of non-pyoverdine producers in iron-limited media that is C-deficient, indicating that they are not exploiting the producers, as would be expected in cheater-cooperator dynamics. But as expected the non-producer has no relative fitness benefit in P-deficient batch culture. In chemostat-like cultures, the non-producer benefits from a relative fitness advantage in the C-deficient media but not the P-deficient media, as predicted by the model.

This is a valuable paper, that shows why it is important to determine the metabolic costs of producing a public good molecule and how this translates into fitness costs, to properly interpret

whether cheater-cooperator dynamics are in fact occurring in the relevant environments studied.

Comments:

-The authors argue that with the batch culture approach alterations in the media may result from "boom-bust" growth patterns, and this may be why the PVD non-producers are not increasing in frequency as expected when in co-culture. To clarify the discrepancy of their results with the large literature of published studies, and to better interpret the results, a few experimental controls/clarifications may be useful:

- 1) what are the growth curves of the strains in the different treatments over 24 hours?
- 2) what are the pyoverdinin production levels (RFU measurements) of the monocultures in the iron-limited C-MEM and P-MEM batch culture treatments? It would be useful to compare them between different concentrations of glucose and phosphate within each treatment type and across each treatment. What is the relationship between OD and PVD production?
- 3) In addition to growing the cultures in iron-limited conditions, it is useful to also compare their growth in iron-rich conditions, where free iron is available to use for the C-MEM and P-MEM treatments. Do PVD non-producers grow as well or better than producers when iron chelation is not required for growth? Or are growth differences due to some other reasons in the media?

-In the batch cultures, why do the non-producers grow better than producers in the P-MEM treatment than the C-MEM treatment while the same is not true for producers? Why is biomass of producers not lower in C-MEM than P-MEM if PVD production costs incurred in C-MEM is expected to have an inverse relationship to biomass. And why is the phosphate concentration that is held constant in the C-MEM treatment much higher (10mM) than used in either of the P-MEM treatments (where higher concentration is 400uM)?

-The chemostat-like co-cultures are difficult to interpret when they cannot be compared to monoculture growths and PVD measurements. Is data available for chemostat monocultures in P-MEM and C-MEM? Would the overall maximum density of the non-producer monoculture be lower than the producer because it has no PVD required for growth? Is the overall density of producer in the C-MEM treatment lower than that in the P-MEM treatment because the metabolic costs of PVD production are higher when C is limited than when P is limited, therefore overall biomass is lower?

Referee #1:

General comment: *“In this manuscript, Sexton and Schuster investigate, from a metabolic perspective, the cost of a secreted “common goods” molecule (an iron scavenging siderophore) by P. aeruginosa. For this, they use a genome-scale metabolic modeling approach followed by chemostat cultivations (allowing constant growth conditions with a single limiting nutrient) with a wild-type strain and a siderophore production deficient strain. Overall, they conclude that the metabolic cost of the siderophore production is dependent on the limiting nutrient (or rather excess availability of specific nutrients) and this ultimately can regulate the emergence and stability of the cheater population. They discuss this conclusion in an ecological context of P. aeruginosa. The conclusion is of general interest. Although the study is sound on technical side, there are several missing links in the chain connecting the data to conclusions.*

Response: Referee #1 presents a number of valuable points that we try to address as thoroughly as possible. In particular, we believe the suggestions to more quantitatively link our empirical and modelling data have strengthened our manuscript. One broad perspective that is apparent in multiple comments that we would like to address more generally is the notion that several of our findings should have been obvious or otherwise expected. Specifically, the reviewer believes the relationship between fitness cost and limiting nutrient should be presented as an obvious finding, suggesting no alternative outcome would be expected. We note while it may have been obvious to the reviewer, it has clearly not been to the field of researchers studying this model system. Since a landmark paper in 2004 (Griffin *et al.*, *Nature*. 2004), dozens of papers have been published from numerous groups studying siderophore cooperation. The relationship between fitness cost and type of nutrient limitation has been completely overlooked, suggesting it has not been obvious.

In the field of microbiology in general, it is common to equate fitness costs with metabolic investment in terms of ATP (as in the textbook classic by Neidhardt *et al.* *Physiology of the bacterial cell*, Sinauer 1990) regardless of specific nutritional context.

Specific point 1: *“The title is misleading, as the study does not address “evolutionary stability”. In fact, the chemostats have not been run long enough to see evolution (e.g. spontaneous emergence of cheaters through mutations). The study is still interesting but the title and other related phrases should be rewritten to avoid misunderstanding.”*

Response: In acknowledgement of the referee’s point regarding the length of our experiments and references to evolutionary stability, we have changed the title of the manuscript to “Nutrient limitation determines the fitness of cheaters in bacterial siderophore cooperation”. We have also modified other related phrasing in the manuscript on page 2, line 3 and page 4, line 17). We would like to point out, however, that it is not uncommon to refer to co-culturing experiments of this type as “experimental evolution”. For example, see Griffin *et al.* *Nature* 2004 or Diggle *et al.* *Nature* 2007.

Specific point 2: *“The link between the modeling and the experiments is there but it is not quantitative. The qualitative hypothesis that the cost is high when substrates needed for the siderophore production are limiting is obvious and could be easily put forward / proved without any modeling. The value of the model would lie in predicting the actual cost in a quantitative manner. This cost is not tested/matched against the experimental observations.”*

Response: As referenced in our general statements, the relationship between limiting substrate and cost has not been as obvious to others in the field as it has been to this referee.

Furthermore, we found that the qualitative model was very helpful to our study as it formalized our hypothesis and streamlined our experimental efforts, demonstrating a value to us without being quantitative. However, we do appreciate the author's point that more can be gained through additional steps to make the model quantitative. We have therefore performed the following analysis. We have experimentally determined the specific growth rates of the WT and the *pvdS* mutant subpopulations in the C-limited chemostat. As opposed to the relative fitness calculations in Fig. 3 that consider average growth rates in batch and chemostat cultures over the entire duration of the culturing period, we have restricted this calculation to the early *pvdS* mutant enrichment phase in the chemostat (up to 72 h of cultivation), before growth rates change and an apparent equilibrium is reached. We have also determined a PVD secretion rate for the WT subpopulation during this period. We used linear and non-linear regression with basic growth and secretion models as described in a new section within the Methods (page 19, lines 5 – 22). Using the experimentally determined values, we interrogated our metabolic model on the impact of secretion on growth, and we found that there is excellent agreement between model and experiment. This finding has been incorporated into the results section (page 9, lines 13-20)

Specific point 3: *“The ‘surprising’ result of high cheater invasion (Fig 4B) is not so surprising but rather ‘interesting’. One of course cannot observe what is infeasible! Again, the authors should be able to show this by using a simple two-component population model. The value here would be if the model could predict this stability point. (Similar arguments would apply to the comments on line 16-17 on page 8).”*

Response: References to the cheater invasion shown in Fig. 4B have been rephrased throughout the text to be presented as “interesting” rather than “surprising”, including page 9, line 6. However, this reviewer appears to be questioning the possibility of a population collapse or “tragedy of the commons” from cheater invasion. Such a population collapse has been observed experimentally (e.g. Dandekar *et al.* *Science* 2012, now added as additional citation on page 9, line 8), and is predicted by social evolution theory: If cooperative cells shared all of the public goods that they created and if the public good provided equal benefit to both the cooperators and the cheaters, yet only the cooperators would bear the metabolic cost of public good production, then cheaters would always out-grow the cooperators. This interaction would be what is called a prisoner's dilemma, in which cooperation is not sustainable in a well-mixed environment (Axelrod and Hamilton, *Science* 1981; Doebeli and Hauert, *Ecol. Lett.* 2005). For co-existence to occur, additional conditions have to be met: For example, in a study that used both chemostat and batch culture approaches, Gore *et al.* (*Nature* 2009) found that a tragedy of the commons in invertase-producing yeast is averted only because producers have preferential access to the sugar produced by invertase. They capture a small portion of the sugar before it diffuses away and is available to all cells.

We would also like to comment on the utility of a two-component population model, which we were initially interested in ourselves. We began to build a simple dynamic cooperator/cheater model as we developed the idea of a chemostat co-culture system, in collaboration with Steffen Klamt from the Max Planck Institute in Magdeburg (Germany). Initially, we found the dynamic model attractive as it predicted an equilibrium of cooperators and cheaters consistent with our experimental data when nutrient-dependent effects are considered, with an explicit function enabling a switch from C-limited to Fe-limited when PVD concentrations decrease below a threshold (without this explicit function, we can observe

cheater invasion to the point of population collapse). However, we found the nature of this equilibrium, and the model in general, to be sensitive to a number parameter values not yet experimentally determined. This led us to decide against such an approach for this study, as the model would not be as robust as we would prefer. Considering that the key mechanistic insight comes from the FBA with the whole-genome metabolic model, we decided it was the most appropriate approach for this manuscript.

Specific point 4: *“Page 10 line 5: hypothesis of C and N excess is easily testable in chemostats and should be done to pin down what is happening.”*

Response: Here the reviewer has requested that we perform additional tests to explore one of the suggested explanations for the producer/non-producer equilibrium observed in Fig. 4B. While we agree further experimentation on this topic would be interesting, we believe the ease at which it can be tested has been underestimated. Chemostat experiments are in fact very labor-intensive compared to the more common batch culturing approaches that have dominated this field. In addition to the duration of the actual culturing, significant preparatory work is required. It would in fact be very challenging to complete such tests within the 3 month window allotted to revisions by the journal. Because the hypothesis referenced by the reviewer was provided as speculated explanation for an observation not central to the main message of the paper, we would like to respectfully make the case that this particular request is beyond the scope of this study. Furthermore, there are other plausible interpretations which also warrant empirical investigation, leading us to believe this question should be more comprehensively addressed in a separate manuscript.

Specific point 5: *“Page 7 line 10-11: Here I see again a conceptual problem. The proportion of cheaters will of course never be more than what the non-cheaters can support. So this comment on inconsistency is violating the fundamental theory of selection.”*

Response: We believe there has been a misunderstanding due to wording choice on our part in the manuscript. We were not trying to make the case that the proportion of cheaters could increase indefinitely, but rather that a cheater is expected to have a relative growth advantage in co-culture with the producer, at least until a threshold ratio of cheaters is reached. This relative growth advantage is essential for the premise that social cheaters could evolve in a natural population. The expectation that a cheater should have a relative fitness advantage (i.e. grow faster than a cooperator) is entirely consistent with evolutionary theory and has been demonstrated in many experimental systems (e.g. Sandoz *et al.* PNAS 2007; Diggle *et al.* Nature 2007; Greig and Travisano, *Proc. Biol. Sci.* 2004; Velicer *et al.* Nature 2000). There is a large body of literature focused on identifying the mechanisms and selective forces which act to prevent a tragedy of the commons outcome. We have hopefully clarified this important distinction by rephrasing Page 7, line 22 – page 8, line 2 to: *“Collectively, we found that PVD non-producers grow at equal rates, but never faster, than the WT. This finding is inconsistent with the notion that spontaneous non-producer mutants could invade a producing population by avoiding the cost of cooperation, at least under the conditions reported thus far.”*

As pointed out above, an equilibrium between cooperators and cheaters is not necessarily the expected outcome. If we assume that cheaters always grow faster than cooperators, and that cooperators produce a public good in a well-mixed environment that allows equal access to both cooperator and cheater subpopulations, then we would expect a

“tragedy of the commons” where cheaters enrich to the point where cooperation can no longer be sustained.

Specific point 6: *“Abstract line 1: “but are generally considered costly for the individual”. By definition, cooperation should cost individual. The question is never about cost but the benefit/cost ratio.”*

Response: We have removed the term “generally” from the abstract on page 2, line 2. Our original choice of words was meant to incorporate the finding from our study that fitness costs are nutritionally conditional. We believe that the reworded statement “...but are considered costly for the individual...” sufficiently conveys this point, and may eliminate a potential source of confusion for a reader in the very first sentence of the abstract. While we agree that the net benefit (benefit minus cost) ultimately determines fitness, we have shown here that cooperation (siderophore secretion) can be essentially “free” if the nutritional conditions are right.

Specific point 7: *“Page 5 line 23: “we found..” should actually be ‘we found, AS EXPECTED, ...’ since there is nothing else that one could expect.”*

Response: We again note while it may have been obvious to the reviewer, it has clearly not been to the field of researchers studying this model system. Despite the fundamental role we demonstrate here in this manuscript, the relationship between fitness cost and growth-limiting nutrient has been completely overlooked, suggesting it has not been obvious. We therefore feel such changes might convey an undesired tone of arrogance to many others in the field to whom it has not been obvious.

More generally, we believe that the impact of nutrient limitation on fitness costs is new and unexpected to most microbiologists, as the central currency to measure cost is considered to be ATP. Furthermore, the observation that PVD production does NOT impose a fitness cost under P-limitation is also not entirely expected, since it could conceivably impose a cost via the P in ATP synthesis (as we point out in the Discussion on page 10, lines 17-21).

Specific point 8: *“Page 9 line 22: PVD production will not have effect on rate in a chemostat but will have effect on yield.”*

Response: We agree that PVD production will have an effect on growth yield during steady-state growth conditions. We did not make it clear enough that we were referring to the relative growth rate differences between PVD-producer and non-producer during the enrichment phase, before a steady-state is reached. While the population growth rate in the chemostat is determined by the dilution rate of the growth medium, variants with decreased investment in secretion (or increased nutrient uptake, to name a different example) will have a higher growth rate that allows invasion and possibly displacement of the resident population (e.g. Zive *et al.* *J. Vis. Exp.* 2013). We have reworded the respective statement on page 10, lines 20-21.

Specific point 9: *“Discussion page 13 line 1: the data does not show (or there is no reference to other literature) whether it is regulatory network acting to shift the siderophore production levels (e.g. by altering gene expression) or it is the metabolic flux/kinetic control subject to intracellular metabolite levels.”*

Response: The statement cited by this referee (now on page 13, line 23 – page 14, line 6) is referring to a role of nutrient-dependent regulation of secretions in *other systems*, for which we

provide the appropriate references (Xavier *et al.*, 2011, Mellbye *et al.*, 2013). With respect to our own study, the results presented in Fig. 4C clearly show that differences in the fitness costs under C vs. P limitation are NOT due to differences in the regulation of Pvd expression. PVD production levels are indistinguishable between P and C-limiting conditions.

Referee #2:

General comment: *“The central question addressed is how nutrient limitation affects cooperative behavior in a microbial community. Addressing this question is of tremendous significance for fundamental evolutionary theory, with applications in a variety of areas.*

The paper capitalizes on the use of mathematical modeling and controlled in vitro evolution experiments to address this complex, systems-level question.”

Specific point 1: *“It’d be interesting to make more use of the model in answering some of the questions posed in the manuscript. Which metabolic pathways are potentially active in the various conditions?”*

Response: We agree! We interrogated our metabolic model to obtain insight into the metabolic basis for the trade-off in growth vs. secretion under C and N-limiting conditions. We compared metabolic fluxes in the presence and in the absence of PVD secretion. We found that amino acid biosynthesis pathways that provide the building blocks for the PVD peptide show the highest increases in flux under PVD secretion conditions. When C or N are growth-rate limiting, then this increased flux depletes precursor metabolites necessary for cellular biomass synthesis (page 6, lines 4-7 in the Results and page 15, lines 7-10 in the Materials and Methods).

Specific point 2: *“Might the model be used to understand why there is a high cheater tolerance in the c-limited chemostat without reducing the total cell density?”*

Response: The metabolic modeling approach we have used is not dynamic and does not capture the interactions between subpopulations over time. Nevertheless, the model suggests that an equilibrium between producers and cheaters can occur when the growth limiting nutrient is something other than C or N. It is therefore plausible that the metabolic activities of the co-culture are leading another growth limiting nutrient to become depleted before carbon in the C-MEM chemostat. In the discussion, (Page 11, lines 2-4), we speculate this could specifically occur: As the WT frequency and hence the total concentration of PVD decreases, iron may become in fact growth limiting. This property would explain the high cheater tolerance (Fig. 4B). However, while we feel it is important to present this explanation, we would like to limit its emphasis without further verification and empirical consideration of other viable alternative explanations referenced in the discussion.

Specific point 3: *“Could the model inform why the pvd mutant has a fitness advantage in the c-mem vs p-mem?”*

Response: The model is indeed oriented around the effects of nutrient limitation and PVD secretion on the growth rate of the WT, and does not explicitly consider co-cultures. However, the general finding that PVD production comes at a fitness cost when a building block of PVD (C and N) is growth rate-limiting, but not when other nutrients are limiting, applies to what might

be expected in co-culture. The model therefore does provide insight into why the PVD mutant has a fitness advantage in C-MEM but not P-MEM, as it shows the WT will grow slower in C-MEM, where the PVD mutant does not. To better highlight the implications of our model to our expectations in co-culture, we have included the following statements: *“This relationship between nutrient limitation and fitness costs has intriguing implications for the evolutionary stability of siderophore secretion by affecting the dynamics between siderophore producers (cooperators) and non-producers (cheaters) in a mixed population.”* (Page 6, lines 9-11)... *“According to our model, PVD production is most costly when a building block of the siderophore is growth limiting (Fig. 1). Hence, a non-producer should enjoy the greatest relative fitness benefit under these conditions.* (page 6, lines 15-17).

Specific point 4: *“Related to the point above, the “parsimonious” explanation with respect to the chemical composition of PVD could be elaborated on more directly.”*

Response: This statement was originally intended to convey that relating the chemical composition of PVD to the growth limiting nutrient provides the simplest explanation for our chemostat results. Because this point is thoroughly elaborated on in the following sentences, it seems the statement referenced by the reviewer has created more confusion than clarity. We have therefore removed entirely the specific wording referenced by the reviewer (Page 10, line 6).

Specific point 5: *“Are there alternative pathways for which biomass or PVD production can be maintained in a more rich media but which are “unavailable” in the c-limited or n-limited media?”*

Response: This question may refer to iron uptake pathways specifically or anabolic pathways in general. We address both here. Although it is known that Pseudomonads have a number of iron uptake mechanisms, it is well established that PVD is specifically needed to acquire iron from a strong chelator such as EDDHA (Cornelis *et al.*, *J. Gen. Microbiol.* 1992). By using a genetically defined *pvdS* mutant (and *pvdD pchEF* mutants in supporting materials), we have disabled the PVD production mechanism without disabling other iron acquisition pathways, demonstrating these other pathways are not involved under our conditions. There is also an extensive body of literature using richer media (KB and CAA) for such co-culturing studies, which are referenced in the discussion.

With respect to anabolic pathways in general, we can state that our minimal medium contains all the building blocks necessary for growth and secretion. In rich medium, cells will likely take up the pre-formed amino acids to satisfy the demand for biomass and PVD synthesis, whereas in minimal medium, they will have to synthesize amino acids *de novo*.

Nevertheless, in the supporting information we explore these issues ourselves using CAA as a richer growth medium. We largely find the results to be similar to those in the C and N limited media, meeting essentially the most stringent standards available to attribute our observations to mechanisms involving PVD, rather than other iron uptake pathways or other anabolic pathways.

Specific point 6: *“There are several statements made in the manuscript that would benefit from a citation (e.g., from the discussion “an established principle...that cells tightly couple biomass yield with ATP generation when limited by the C and energy source”).”*

Response: Good point. We have included several additional references throughout the manuscript, including the one specifically mentioned by this referee. The reference in question (Russell and Cook, 1995) is identical to that in the subsequent sentence. For clarification, we have explicitly added it to the preceding sentence as well (page 10, lines 13-14).

Specific point 7: *“The methods state that “growth rate” is calculated and compared to the PVD secretion rate. How are these rates normalized?”*

Response: We have indeed normalized growth and secretion rates for visualization in Fig. 1B. We have added a statement on page 14, lines 22 – page 15, line 1 that describes the normalization process.

Specific point 8: *“The methods state that the “C,N,P,S, and Fe uptake rates were set such that the resulting growth rate in the absence of any PVD secretion is identical to the experimental value”...but wouldn’t the relative balance between the C,N,P,S, and Fe uptake rates affect some of the analyses comparing growth and PVD secretion rate? Perhaps an analysis varying the relative uptake rates of those inputs should be done to verify the observations of the balance between growth and PVD secretion.”*

Response: We agree that we were not very clear on how the various uptake rates are set. We have improved our methods description on page 14, lines 17-22. In short, we always limit the uptake rate of the one nutrient in question, and keep all other nutrient uptake rates either at the maximum experimentally determined value (in case of C) or essentially unbounded (in case of N, P, S, and Fe).

Inspired by this reviewer’s comment, we have also performed a systematic analysis of the trade-off between growth and secretion by varying C and P uptake rates (Supplementary Fig.1, page 6, line 1-3 in the Results, and page 15, lines 3-6 in the Methods). We show that it is the relative availability of C vs. P that determines the magnitude of the trade off in terms of a reduction in growth rate.

Specific point 9: *“How are the “C,N,P,S, and Fe” uptake rates actually constrained? It looks like “C” is really defined as glucose...but do the results change when “C” is some other carbon source?”*

Response: The referee’s assumption is correct in that C is defined as glucose. We have taken the opportunity to also investigated alternative C sources relevant to *P. aeruginosa* growth physiology: an amino acid (glutamate) and a TCA-cycle organic acid (succinate). We find that all three C-sources produce identical results (provided their respective uptake rates are set such that they result in identical growth rates in the absence of PVD secretion). We report this result on page 5, line 19-20 and describe the methods on page 15, lines 1-2. Because lines for glucose, succinate, and glutamate would completely overlap in Fig. 1B, we found it unnecessary to include a separate graph for these cases.

Specific point 10: *“Fig 3 would benefit from a title above A and B to readily distinguish between batch and chemostat.”*

Response: We agree that titles above the individual panels of Fig. 3 improve clarity and have followed the referee’s suggestion.

Referee #3:

General comment: *“The authors study the underlying costs of producing the well-studied public good molecule, the iron-chelating pyoverdine siderophore, produced by Pseudomonas aeruginosa bacteria. They show how production costs vary in different environmental conditions and determine the implications this has on the evolutionary stability of cooperation. Using metabolic modelling and experiments, they show that although the production of pyoverdine is always energetically costly, it only has a fitness cost when the essential nutrients required to make it (carbon and nitrogen) are limiting. The metabolic model shows that there is an inverse relationship between pyoverdine production and cell biomass only when C and N are limited but not when Fe, P or S are limited. In contrast to predictions made by the model, batch culture experiments show that there is no relative growth advantage of non-pyoverdine producers in iron-limited media that is C-deficient, indicating that they are not exploiting the producers, as would be expected in cheater-cooperator dynamics. But as expected the non-producer has no relative fitness benefit in P-deficient batch culture. In chemostat-like cultures, the non-producer benefits from a relative fitness advantage in the C-deficient media but not the P-deficient media, as predicted by the model.*

This is a valuable paper, that shows why it is important to determine the metabolic costs of producing a public good molecule and how this translates into fitness costs, to properly interpret whether cheater-cooperator dynamics are in fact occurring in the relevant environments studied.”

Specific comments

Specific point 1: *“The authors argue that with the batch culture approach alterations in the media may result from “boom-bust” growth patterns, and this may be why the PVD non-producers are not increasing in frequency as expected when in co-culture. To clarify the discrepancy of their results with the large literature of published studies, and to better interpret the results, a few experimental controls/clarifications may be useful”:*

Response: The referee has asked us to elaborate on the nature of growth in batch culture, and to specifically relate our findings in relation to previous literature. In the points below, we took the effort to thoroughly address referee #3’s individual questions. We would like to point out, however, that significant attention in the discussion of the original manuscript has already been dedicated to this end. A segment in the Discussion on pages 11-12 is dedicated to this topic, along with an extensive data set of controlled biologically independent experiments in the supporting information.

Specific point 2: *“What are the growth curves of the strains in the different treatments over 24 hours?”*

Response: In our initial preliminary investigations, we had performed such growth curves using microtiter 96 well plates, which provide qualitatively representative data. However, these experiments were not maintained for 24 hours due to the significant impact of evaporation in small volumes, which is why they were not presented originally. To acquire data across the 24 hours as requested by the referee, we have performed a new set of growth curves and present the results in what is now the new Supplementary Fig. 2A. This data is also referenced in the results section of the main text (page 7, lines 6-8).

Specific point 3: *“What are the pyoverdinin production levels (RFU measurements) of the monocultures in the iron-limited C-MEM and P-MEM batch culture treatments? It would be useful to compare them between different concentrations of glucose and phosphate within each treatment type and across each treatment. What is the relationship between OD and PVD production?”*

Response: This request was considered when we performed the growth curves to address the previous comment. We therefore incorporated RFU measurements to the sampling scheme and this correlating RFU data is provided as panel B of the new Supplementary Fig. 2, referenced in the main text on page 7, line 11.

Specific point 4: *“In addition to growing the cultures in iron-limited conditions, it is useful to also compare their growth in iron-rich conditions, where free iron is available to use for the C-MEM and P-MEM treatments. Do PVD non-producers grow as well or better than producers when iron chelation is not required for growth? Or are growth differences due to some other reasons in the media?”*

Response: The referee brings up an important point regarding our ability to attribute the phenotype of the *pvdS* mutant specifically to iron limitation, rather than any other growth condition. We can indeed confirm that adding free iron restores growth of the PVD-non producers in MEM. While this was not explicitly reported as results in the original manuscript, this characteristic was in fact fundamental to our experiment design; before co-culturing, the WT and *pvdS* mutant were pre-cultured separately in an iron-replete MEM, where the mutant is able to grow independently. We chose this medium specifically because it permits robust growth of both WT and *pvdS* mutant, while retaining all other shared properties of our experimental media. This step is described in the materials and methods (page 17, lines 2-6), but was not made clear when the results are presented. To emphasize this important point, we have included a new figure (Supplementary Fig. 3) which shows equivalent growth of both strains in iron replete MEM. This point is referenced in the results section (page 7, lines 11-13)

Specific point 5: *“In the batch cultures, why do the non-producers grow better than producers in the P-MEM treatment than the C-MEM treatment while the same is not true for producers?”*

Response: We appreciate the referee drawing our attention to this detail in the original Fig. 2. Although the *pvdS* mutant is clearly handicapped compared to the WT, slightly higher levels of mutant growth can be discerned in P-MEM than in C-MEM. We investigated this and were surprised to find that while diligent in using fresh preparations of EDDHA in all other experiments presented throughout the manuscript, our records show an older chelator preparation had accidentally been used for the P-MEM batch culture experiment (but not chemostat). A plausible explanation is that partial degradation of EDDHA could have made some iron freely available, manifesting as the marginal growth observed by the mutant, while having little impact on the WT which can access all the iron whether or not the chelator is functional due to PVD production. To confirm this explanation, we have repeated the experiment with a fresher EDDHA prep. As expected, we found that the fresher chelator prep completely prevented growth of the *pvdS* mutant (see updated Fig 2, and Supporting Fig. 2A).

Specific point 6: *“Why is biomass of producers not lower in C-MEM than P-MEM if PVD production costs incurred in C-MEM is expected to have an inverse relationship to biomass?”*

Response: The media formulations were specifically designed such that both P-MEM and C-MEM would sustain equal cell densities (now made clear on page 7, line 6-8). As shown in Fig. 2, this was done by selecting dilutions of the limiting nutrient that yielded equal densities. This approach therefore accounts for any difference in production costs.

Specific point 7: *“And why is the phosphate concentration that is held constant in the C-MEM treatment much higher (10mM) than used in either of the P-MEM treatments (where higher concentration is 400uM)?”*

Response: 10 mM phosphate is the standard concentration used in Evan’s minimal medium (MEM) for bacterial growth. However, to limit P such that the optical density is still equal to 1, we had to decrease the concentration to 200 μ M. Note that 400 μ M P was not used in any other experiment than that presented in Fig. 2, where it served a distinct purpose of demonstrating P limitation, as doubling from 200 to 400 μ M led to an approximate doubling in culture density.

Specific point 8: *“The chemostat-like co-cultures are difficult to interpret when they cannot be compared to monoculture growths and PVD measurements. Is data available for chemostat monocultures in P-MEM and C-MEM?”*

Response: We believe that our co-culture chemostat results are quite clear and easy to interpret, because we compare two otherwise identical conditions that vary only in a single parameter (the growth-limiting nutrient), leading us to make a straightforward conclusion about nutrient-dependent fitness costs that is consistent with our metabolic model. In any case, monoculture chemostats are not possible with mutant alone, as PVD production is required for growth. Considering the intensive nature of chemostat operation, we have not been compelled to empirically demonstrate the wash-out event that would occur.

Specific point 9: *“Would the overall maximum density of the non-producer monoculture be lower than the producer because it has no PVD required for growth?”*

Response: Yes! As indicated in the preceding response, the inability of the *pvdS* mutant to grow and to produce PVD under Fe-stringent conditions does not allow it to reach sufficient density in batch culture, let alone in a chemostat in which the culture is constantly diluted with fresh medium. Thus, chemostat operation would lead to immediate wash-out of the mutant strain.

Specific point 10: *“Is the overall density of producer in the C-MEM treatment lower than that in the P-MEM treatment because the metabolic costs of PVD production are higher when C is limited than when P is limited, therefore overall biomass is lower?”*

Response: This comment is similar to one made earlier (point #6). The overall cell densities are a result of the deliberate formulation of the growth media (C-MEM and P-MEM, respectively). It was our intent to keep cell densities very similar to eliminate differences as a potential confounding factor. However, we agree with the reviewer’s general interpretation on the effect of nutrient limitation on growth. In principle, the concentration of the growth-rate limiting nutrient determines the growth yield (overall density) in the chemostat at steady-state. Thus the trade-off between growth and secretion under C-limitation would divert some of the available C into PVD secretion, resulting in lower overall cellular biomass production and hence, density.

Reviewers' comments:

Reviewer #1 (Remarks to the Author):

The authors have addressed many of my comments satisfactorily. There are a few points that I think will be important before the manuscript is accepted for publication.

Major comments:

1. A two-component population model seems necessary to make the message clearer. Even if qualitative, such model would help understanding the dynamics of two populations better.
2. Regarding the authors' response to my previous comments, there are two things that I recommend to the authors to check (in terms of phrasing etc.) while preparing the next version. The authors seem to believe that a chemostat can support different growth rate populations. This can happen only till the steady state is reached (or when the steady state is dynamic, e.g. oscillatory). The invasion of cheaters and subsequent population collapse is of course a possibility but that depends a lot on the nature of dependence (of cheaters). In case the relation is obligate, one will observe a convergence towards a stable state (or oscillations as in a prey-predator system). Of course, stochasticity and spontaneous mutation rates etc. would alter the picture.

Minor comments:

1. Abstract, line 33: "an unconventional continuous-culture...". I do not see the necessity of highlighting "unconventional" here. It does not say anything about the biological justification of the choice of the system. On the other hand, it may not be received well by many applied microbiologists who have been using chemostats for decades (after all, the literature volume based on chemostat cultures is not small by any means).
2. Line 115: "depletes the precursor metabolites..". Steady-state models can't say anything about metabolite levels but only about flux redistributions.
3. Figure 2 caption: I guess the glucose and P concentrations refer to initial concentrations; if not, how were they held constant?

Reviewer #2 (Remarks to the Author):

The authors thoroughly addressed my comments/suggestions. I have no further comments.

Reviewer #3 (Remarks to the Author):

I am satisfied with the authors response to reviewers.

Author response to referees' comments for manuscript NCOMMS-16-25498A

Referee #1:

General comment: *"The authors have addressed many of my comments satisfactorily. There are a few points that I think will be important before the manuscript is accepted for publication."*

Response: We thank this reviewer for the thorough review of our manuscript, and we appreciate the opportunity to again address each of the points raised in detail.

Major comment 1: *"A two-component population model seems necessary to make the message clearer. Even if qualitative, such model would help understanding the dynamics of two populations better."*

Response: We appreciate this reviewer's request. However, in weighing the potential benefit versus the substantial effort involved, we come to the conclusion that a two-component (dynamic) population model is not appropriate for the scope of this study. We are convinced that this model would add little to the main conclusions of the paper. As we indicated in our last response, different dynamic modeling outcomes are possible depending on the parameters chosen. Many parameters have not been empirically determined, and it would be beyond the scope of this study to do so. We believe that it would add uncertainty and dilute rather than affirm the main message that we hope to send to a broad audience.

The main conclusion of our paper is that the costs of secretions are dependent on the nature of the growth-rate limiting nutrient, and that this relationship impacts the stability of cooperative behavior. Our metabolic model, together with careful experimentation, provides evidence in full support of this conclusion. All three reviewers agree this is the case. Our conclusion is significant and novel. It has the potential to transform the current thinking in the field of microbial social evolution, and it challenges a series of experimental studies that have been performed in batch rather than chemostat format.

Simulations with a dynamic model would not add anything to this main conclusion. They might help explain the stable co-existence of cooperators and cheaters we observed in C-limited chemostats, an interesting yet separate question that emerged as a by-product of our main finding. However, a *qualitative* model insensitive to potentially important experimental parameters would not provide a satisfactory answer to this question. In our opinion, a meaningful, quantitative dynamic model would build on the well-established differential equation framework for the chemostat and would capture intricate nutrient-dependent effects on growth. It would identify possible stable states of the system and their dependence on (or independence of) parameter space. This would require a mathematical depth that we feel is inappropriate for the focus of this manuscript.

Major comment 2: *"Regarding the authors' response to my previous comments, there are two things that I recommend to the authors to check (in terms of phrasing etc.) while preparing the next version. The authors seem to believe that a chemostat can support different growth rate populations. This can happen only till the steady state is reached (or when the steady state is dynamic, e.g. oscillatory). The invasion of cheaters and subsequent population collapse is of course a possibility but that depends a lot on the nature of dependence (of cheaters). In case the relation is obligate, one will observe a convergence towards a stable state (or oscillations as in a prey-predator system). Of course, stochasticity and spontaneous mutation rates etc. would alter the picture."*

Response: We fully agree with the reviewer's statement that subpopulations with

different growth rates are maintained only until a steady-state is reached. We have attempted to make this clear in the last revision. To further improve clarity on possible steady states, we have reduced the emphasis on population collapse in the context of our chemostat system by removing the relevant statement in the Results section (page 9, lines 186-187). We now discuss population collapse or the “tragedy of the commons” as the default outcome of obligate cheating in more general terms in the discussion, citing empirical and experimental evidence. This de-emphasizes the direct link between the tragedy of the commons and our experimental system, which may have been a point of contention.

We maintain our general position, however, that it is a theoretically and empirically supported fact that obligate cheating is expected to cause a tragedy of the commons or population collapse (citations in the revised manuscript). Because cheaters save the costs of cooperation, they will inevitably increase in frequency until cooperation is no longer possible. Here, the only time that subpopulation growth rates will NOT be different (the steady-state referred to by the reviewer) is when the population has collapsed and growth rates are zero.

As our experimental data in the manuscript suggest an equilibrium of subpopulations rather than a population collapse, we feel that it is justified to emphasize this unexpected observation, just as others did in different experimental systems (e.g. Dandekar et al. Science 2012; Gore et al. Nature 2009). We offer several mechanistic explanations in the Discussion of how such an equilibrium can be achieved. Importantly however, whether one accepts the tragedy of the commons as the default outcome of obligate cheating or not has no bearing on the main conclusions of our manuscript. The observation that the fitness costs and the stability of cooperation are nutrient-dependent stands regardless.

Minor comment 1: *“Abstract, line 33: “an unconventional continuous-culture...”. I do not see the necessity of highlighting “unconventional” here. It does not say anything about the biological justification of the choice of the system. On the other hand, it may not be received well by many applied microbiologists who have been using chemostats for decades (after all, the literature volume based on chemostat cultures is not small by any means).”*

Response: We agree with the reviewer that the general use of chemostats in microbiology is by no means unconventional. We used this adjective to refer to the rare use of chemostats in the field of social microbiology, and specifically in siderophore cooperation. Nevertheless, we have modified this statement to provide a more direct, biological justification of the choice of the system (page 2, line 33): *“We confirm these results experimentally with a continuous-culture approach that allows precise adjustment of the limiting nutrient.”*

Minor comment 2. Line 115: *“depletes the precursor metabolites..”. Steady-state models can’t say anything about metabolite levels but only about flux redistributions.”*

Response: Good point. Our statement was an interpretation of the modeling data that indicated the likely consequences of altered reaction fluxes. To merely report modeling outcomes, we have rephrased the statement on page 6, line 115 as follows: *“Under C and N-limitation, this increase in turn restricts reaction fluxes from central precursor metabolites to cellular biomass”.*

Minor comment 3. *“Figure 2 caption: I guess the glucose and P concentrations refer to initial concentrations; if not, how were they held constant?”*

Response: Correct, the given concentrations for glucose and phosphate are initial concentrations. We have added a sentence in legend of Fig. 2 to make this point more clear (page 25, line 618): *“The initial concentrations of glucose and phosphate in the medium vary as*

follows.”

Reviewer #2:

General comment: *“The authors thoroughly addressed my comments/suggestions. I have no further comments.”*

Response: We are glad that this reviewer is fully satisfied with our revisions.

Reviewer #3:

General comment: *“I am satisfied with the authors response to reviewers.*

Response: We appreciate the positive response to our revisions!

REVIEWERS' COMMENTS:

Reviewer #1 (Remarks to the Author):

I have no further comments. The manuscript is ok for publication.

REVIEWERS' COMMENTS:

Reviewer #1 (Remarks to the Author):

"I have no further comments. The manuscript is ok for publication."

Author's response:

We appreciated this referee's constructive comments on our manuscript and are glad they find it suitable for publication.